# Endoplasmic Reticulum Stress-Related Ten-Biomarker Risk Classifier for Survival Evaluation in Epithelial Ovarian Cancer and *TRPM2*: A Potential Therapeutic Target of Ovarian Cancer

**DOI:** 10.3390/ijms241814010

**Published:** 2023-09-12

**Authors:** Minghai Zhang, Yingjie Wang, Shilin Xu, Shan Huang, Meixuan Wu, Guangquan Chen, Yu Wang

**Affiliations:** 1Department of Obstetrics and Gynecology, Shanghai First Maternity and Infant Hospital, School of Medicine, Tongji University, Shanghai 200092, China; zhangminghai@renji.com (M.Z.);; 2Department of Obstetrics and Gynecology, Renji Hospital, School of Medicine, Shanghai Jiaotong University, Shanghai 200127, China; 3Shanghai Key Laboratory of Maternal Fetal Medicine, Shanghai Institute of Maternal-Fetal Medicine and Gynecologic Oncology, Clinical and Translational Research Center, Shanghai First Maternity and Infant Hospital, School of Life Sciences and Technology, Tongji University, Shanghai 200092, China

**Keywords:** epithelial ovarian cancer, endoplasmic reticulum stress, risk classifier, *TRPM2*

## Abstract

Epithelial ovarian cancer (EOC) is the most lethal gynecological malignant tumor. Endoplasmic reticulum (ER) stress plays an important role in the malignant behaviors of several tumors. In this study, we established a risk classifier based on 10 differentially expressed genes related to ER stress to evaluate the prognosis of patients and help to develop novel medical decision-making for EOC cases. A total of 378 EOC cases with transcriptome data from the TCGA-OV public dataset were included. Cox regression analysis was used to establish a risk classifier based on 10 ER stress-related genes (ERGs). Then, through a variety of statistical methods, including survival analysis and receiver operating characteristic (ROC) methods, the prediction ability of the proposed classifier was tested and verified. Similar results were confirmed in the GEO cohort. In the immunoassay, the different subgroups showed different penetration levels of immune cells. Finally, we conducted loss-of-function experiments to silence *TRPM2* in the human EOC cell line. We created a 10-ERG risk classifier that displays a powerful capability of survival evaluation for EOC cases, and *TRPM2* could be a potential therapeutic target of ovarian cancer cells.

## 1. Introduction

EOC is the most lethal gynecological malignant tumor and the fifth most common cause of cancer deaths in women [1]. Although continuous improvements in drugs and surgical techniques have emerged in recent years, the prognosis of EOC patients has not improved significantly, mostly because of the characteristics of morbidity concealment and complex pathogenesis [2]. Currently, the most frequently employed biomarker in clinical diagnosis is CA125, which is neither sensitive nor specific enough to reflect early-stage illness and prognosis [3]. Reliable prognosis prediction models are still needed to screen high-risk patients for relatively early clinical intervention.

Recently, ER processes were found to be highly associated with tumor progression [4]. The ER, which is the largest protein processing organelle in cells, can precisely regulate the whole process of protein synthesis and transportation [5]. ER function can be disrupted by numerous abnormal cellular states, which causes ER stress [6]. Because of the accumulation of misfolded/unfolded proteins resulting from ER stress, the unfolded protein response (UPR) is triggered and disrupts the translation of proteins and degrades aggregated misfolded/unfolded proteins. If the protein cannot maintain its homeostasis when disturbed by the UPR, the cell will undergo apoptosis [7]. A growing number of studies have indicated that the regulation of ER stress is closely related to the growth, metastasis, and recurrence of various tumors, and ER stress-related genes (ERGs) play critical roles in tumor progression [8].

Several ER stress-related proteins (such as GRP78, ATF6, and PERK) are highly expressed in EOC patient tissues. For instance, increased protein expression of GRP78 is associated with worse patient outcomes [9]. Recent studies have shown that susceptibility to EOC cell apoptosis is modulated by PHLDA1 and OMA1 via the ER stress response pathway [10,11]. Although an increasing number of studies have indicated that ER stress plays an important role in the pathological processes of malignant tumors, ER stress has not been adequately studied in EOC.

At present, prognostic models of EOC based on multiple biomarkers have attracted increasing attention [12,13]. However, a prognostic classifier based on ERGs has not yet been widely studied in EOC yet. Here, an ER stress-based risk classifier of EOC stemming from a gene matrix was generated and validated in public EOC patient data. The results in this paper are also helpful for understanding the role of ER stress in the genesis and development of EOC. Additionally, transient receptor potential melastatin 2 (*TRPM2*), which was found to be a crucial factor in the ER stress-related risk classifier, was functionally validated in a human EOC cell line. Our proposed classifier tool may benefit personalized treatment for EOC patients, and *TRPM2* could be a promising prognostic biomarker and a potential therapeutic target.

## 2. Results

### 2.1. The Characteristics of the Differentially Expressed ERGs (DEERGs)

A brief flowchart of our study is shown in Figure 1. In total, 7617 significant differentially expressed genes (DEGs) were identified in ovarian tumor samples compared with normal tissues (Figure 2A). After overlapping the DEG and ERG sets, 422 DEERGs were obtained (Figure 2B).

### 2.2. Construction and Verification of the ER Stress-Related Classifier

First, univariate logistic regression was performed, and 61 candidate DEERGs that were significantly associated with overall survival in the TCGA-OV cohort were generated. Then, LASSO regression was used to remove overfitted genes of candidate DEERGs, and 18 genes were obtained (Figure 2C,D). Finally, an ER stress-related classifier including 10 candidate DEERGs, which were screened out by multiple stepwise regression analysis of the 18 genes selected by LASSO, was constructed (Figure 2E). The equation of the classifier was as follows: risk score = 0.1295 × *TRPV4* + 0.1862 × *TRPM2* − 0.3113 × *STAT1* − 0.2665 × *RCN2* − 0.0760 × *MZB1* − 0.1118 × *GPR37* + 0.1284 × *GABARAPL1* + 0.2150 × *FOXO1* + 0.1338 × *CDKN1B* + 0.0914 × *CACNA1C*.

The generated predictive value of the ER stress-related classifier for prognostic outcomes of EOC patients was also investigated. In the TCGA-OV cohort, there were significant differences in the survival curves between the low-risk and high-risk groups, and the patients with high risk scores presented poor prognostic outcomes (Figure 3A). In addition, ROC curves indicated favorable predictive accuracy, in which the AUCs for the one-, three-, five-, and seven-year survival rates were equal to 0.70, 0.66, 0.73, and 0.79, respectively (Figure 3B). After analyzing the relationship between different risk scores, follow-up times, and the expression of 10 candidate DEERGs, the results indicated that with increasing risk scores the survival rate of patients decreased significantly (Figure 3C). Meanwhile, similar outcomes were observed in the verification cohorts (GSE32062 and GSE140082) by the same analysis (Figure 3D,E, Appendix A).

### 2.3. Establishment of a Prognostic Nomogram

After univariate, LASSO, and multivariate COX analyses, the risk score of the ER stress-related classifier was indicated to be a reliable indicator to predict the survival time of EOC patients. Moreover, we selected two important clinical variables (R_0_/non-R_0_ and age) from several clinical characteristics in the TCGA-OV dataset based on the multivariate COX analysis results (Figure 4A). The two selected clinical variables were combined with the risk scores to develop a nomogram that could calculate a value for each patient to predict the survival outcome (Figure 4B). A higher value represented a better survival outcome (Figure 4C). The ROC and calibration curves indicated that the classifier-based nomogram had good reliability (Figure 4D,E).

### 2.4. Exploration of the Relationship between the ER Stress-Related Classifier and Clinical Parameters

We compared differences in risk score levels based on several clinical parameters, such as clinical stage, grade, and tumor type. In the TCGA-OV dataset, no discrepancy was found between the two groups stratified based on tumor grade, except that patients with advanced clinical stages and with primary tumors had higher risk scores (Figure 5A,B).

### 2.5. Analysis of Immune Cell Infiltration in the Two Groups

The immune infiltration levels of the 22 cell types according to the CIBERSORT algorithm were also implemented. The results indicated that the infiltration ratios of memory-activated CD4 T cells, follicular helper T cells, M1 macrophages, and plasma cells in the high-risk score group were much lower than the corresponding infiltration ratios in the low-risk score group. In contrast, the infiltration ratios of M2 macrophages and monocytes in the high-risk score group were higher than the corresponding infiltration ratios in the low-risk score group (Figure 5C). In addition, we checked the expression level of 10 immunosuppressive checkpoints to better understand the differences in the tumor microenvironment (TME) between the two groups. The expressions of *BTLA*, *CD274*, *CTLA4*, *LAG3*, and *TIGIT* were significantly downregulated in the high-risk score group. In contrast, the expression of *CD160* was significantly upregulated in the high-risk score group (Figure 5D).

### 2.6. qPCR Assay of 10 Candidate DEERGs

Primers for 10 candidate DEERGs among the generated ER stress-related classifiers (Table 1) were designed, and the gene expression of the 10 DEERGs was tested in cell lines. The result of the qPCR assay was consistent with previous results, which showed that 10 candidate DEERGs were significantly expressed between normal ovarian epithelial cells and cancer cells (Figure 6).

### 2.7. Knockdown of TRPM2 Inhibited Ovarian Cancer Cell Apoptosis, Invasion, and Migration

Among the 10 candidate DEERGs, a significant association between the expression of *TRPM2* and the survival of ovarian cancer patients in the TCGA database was discovered (Figure 7A). To explore the role of *TRPM2* in the progression of EOC, loss-of-function experiments to silence *TRPM2* in a human EOC cell line were performed. The efficiency of shRNA knockdown was confirmed by qPCR (Figure 7B). Among the selected shRNAs, sh*TRPM2*-3 exhibited the highest level of gene silencing efficiency and was thus chosen for subsequent experiments. The CCK-8 assay revealed that knocking down *TRPM2* had no significant effect on the proliferation of SKOV3 cells (Figure 7C). Compared to the control, downregulation of *TRPM2* resulted in G2/M cell cycle arrest in SKOV3 cells (Figure 7D) and significantly upregulated the apoptosis rate of SKOV3 cells (Figure 7E). Moreover, the findings from scratch assays (Figure 7F) and transwell invasion assays (Figure 7G) provided additional evidence that silencing *TRPM2* significantly reduced the migration and invasion capacities of SKOV3 cells. The downregulation of *TRPM2* levels resulted in an elevation in intracellular reactive oxygen species (ROS) levels, indicating that *TRPM2* may have enhanced ovarian cancer cell activity by inhibiting ROS (Figure 7H). This result is also similar to the conclusions of other studies [14].

## 3. Discussion

EOC is a lethal gynecologic malignancy with dismal patient outcomes due to the complexity of its tumor heterogeneity [15]. To date, several previous studies have made efforts to predict the prognosis of EOC and have made encouraging achievements. However, the predictive abilities of these biomarkers are limited, and more effective and accurate classifiers need to be developed [12,13,16,17,18]. Several experimental studies have shown that ER stress and the unfolded protein response are associated with the occurrence, development, and invasion of EOC [19]. However, few studies have been implemented to illuminate the exact role of ER stress in EOC. To date, few studies have investigated the role of *TRPM2* in ovarian cancer. In this study, we established a novel prognostic classifier of EOC and conducted a preliminary exploration of the potential regulatory role of *TRPM2* in the development and progression of EOC.

Here, we created a novel prognostic classifier based on 10 ER stress-related genes (TRPV4, *TRPM2*, STAT1, RCN2, MZB1, GPR37, GABARAPL1, FOXO1, CDKN1B, and CACNA1C) (Figure 2). The overall survival of the low-risk group was significantly higher than that of the high-risk group. The results (Figure 2 and Figure 3) showed that our new classifier possessed excellent performance in the prognostic prediction of EOC. Li et al. used GEO datasets to validate ferroptosis-related and necroptosis-related prognostic models; however, they focused on survival analysis and lacked further experimental-level validation, such as gene functional analysis [12]. Hu et al. focused on five regulators of G protein signaling genes and constructed a risk score to predict prognosis. The ROC curve of the risk score was not outstanding enough compared to ours [13]. One of the reasons that our model demonstrated excellent performance is that ERs, which are affected by much of the intracellular and environmental damage related to cancer, may play a crucial role in clinical prognosis [20].

Among the prognostic classifiers in this study, the Forkhead box O1 (FOXO1) protein plays an important role in carcinogenesis [21]. The abnormal expression or activity disturbance of the FOXO1 transcription factor is related to multiple gynecological diseases, such as endometrial cancer, endometriosis, and ovarian dysfunction, which in turn highlights its potential as a therapeutic target [22,23]. It has been proven that FOXO1 is associated with resistance to cisplatin and targeted specific drugs [19]. The FOXO1 downstream effector participates in the process of signal transduction [24]. Recently, FOXO1 began to be used as a potential target area for antitumor effects. For example, interfering with FOXO1 localization has been proposed as a possible prospective approach to improving cell sensitivity to cisplatin because nuclear retention of FOXO1 may be beneficial for the induction of proapoptotic genes [19].

Transient receptor potential vanilloid 4 (TRPV4) is a nonselective calcium-permeable cation channel that has been indicated to play many important physiological roles, and various disease states are attributed to the absence or abnormal function of this ion channel [25]. TRPV channels are overexpressed in breast cancer. Ca^2+^ influx mediated by TRPV4 may help to enhance cancer cell proliferation and other important processes in tumor progression, such as angiogenesis [26].

The activator of transcription 1 (STAT1) is a protective factor in our classifier. Several studies have indicated that activation of STAT1 plays a tumor suppressor role in cancer cells [27]. STAT1 is a tumor inhibitor, and its possible mechanism may be related to its subtype, STAT1β. In addition, reticulocalbin 2 (RCN2) is also a protective factor. Reticulocalbin is a kind of Ca^2+^ binding protein, and multiple EF-hand proteins are mainly located in the endoplasmic reticulum. The function of reticulocalbin in tumor progression has not been indicated. Some studies have shown that reticulocalbin may affect tumor progression and platinum resistance by regulating Ca^2+^ homeostasis [28,29].

We divided the EOC patients into two ER stress-related groups, in which different penetration levels of immune cells were investigated (Figure 5). The results revealed that various immune cells, such as macrophages, CD4+ T cells, and T follicular helper (Tfh) cells, were greatly suppressed in the high-risk score group. Macrophages and monocytes are important components in the tumor microenvironment. In our results, monocytes were downregulated and M1 macrophages were upregulated in the low-risk score group. A previous study indicated that monocytes can differentiate into macrophages, which further differentiate into M1 and M2 macrophages under the stimulation of cytokines and chemokines. M1 macrophages release a large number of proinflammatory cytokines, which are conducive to tumoricidal activity and antigen presentation [30]. According to our model, it was indicated that the poor prognosis in the high-score group of patients may be attributed to the downregulation of M1 macrophages, leading to a weakened antitumor effect. In addition, we observed that memory-activated CD4+ T cells and Tfh cells were upregulated in the low-risk group. T–B-cell aggregation and interaction sometimes forms tertiary lymphoid structures (TLSs) in the area around malignant tumors. Tfh cells produce robust CXCL13 23, 24, 27, 44, 55, which may contribute to the formation of TLSs in tumors. The presence of CXCL13-producing CD4+ T cells and the formation of TLSs were associated with improved survival in patients with several malignant tumors [31]. The favorable prognosis in the low-risk group may be attributed to TLS. In addition, we found that suppressive immune checkpoint genes were upregulated in the low-risk score group. These outcomes indicate that ER stress-related processes may play an important role in tumor microenvironment regulation. Tumor immune escape (TIE) is the driving force of tumor development. Immunosuppressive signals are transmitted to immunosuppressive cells through receptor–ligand interactions, restraining the tumoricidal effect [32]; thus, patients with high expression of immune checkpoint genes may benefit from immunotherapy. Based on our analysis, we speculate that patients in low-risk groups tend to benefit from immunotherapy.

Moreover, gene ontology and pathway analysis of 422 DEERGs was performed based on IPA analysis. The results indicated that the DEERGs were closely related to cytoactivity and apoptosis (such as cellular maintenance, cell survival, and organismal abnormalities) (Figure 8). Through IPA analysis, the significantly regulated signaling pathways affected by ERGs were analyzed (Figure 8A). Acute phase response signaling was upregulated, and the ribonucleotide reductase signaling pathway was downregulated. Reticulum stress triggers the UPR, which leads to excessive production of acute phase proteins and inflammation in ovarian tumor tissue. Ribonucleotide reductase (RR) participates in the process of DNA synthesis and repair [33]. ER stress may inhibit the process of DNA synthesis by restraining RR activity and can eventually lead to cell cycle arrest and apoptosis. Moreover, based on the IPA analysis of 422 DEERGs, it was shown that TP53 may be a potential upstream regulatory factor of ER stress in ovarian cancer (Figure 8B). The TP53 tumor suppressor gene is a key factor in regulating cell growth, homeostasis, and survival [34]. The abnormal expression of TP53 results in cell cycle disorder, which leads to a stress phenotype.

Finally, loss-of-function experiments were implemented to investigate the biological function of *TRPM2*, which played an important role in the prognostic classifier (Figure 2E). *TRPM2* is a nonselective Ca^2+^-permeable cation channel that is highly sensitive to the activation of oxidative stress and is widely distributed throughout the body [35]. Recently, several studies have indicated that *TRPM2* plays an important role in mediating cell death induced by miscellaneous oxidative stress-inducing pathological factors [36]. The activation of *TRPM2* channels changes the homeostasis of intracellular ions, leading to abnormal activation of various cell death pathways. *TRPM2* is highly expressed in many cancers, such as breast cancer, prostate cancer, and pancreatic cancer. The *TRPM2* protein has also been proven to maintain the viability of tumor cells in various cancers. Activation of *TRPM2* results in the expression of transcription factors and kinases that are critical for tumor cell proliferation and survival, such as CREB, HIF-1/2α, Nrf2, and Src phosphorylation [37]. The results in this study also confirmed a similar conclusion in ovarian cancer (Figure 7). For instance, *TRPM2* was significantly upregulated in ovarian cancer cells compared to normal cells, and *TRPM2* promoted intracellular ROS and the invasion and migration of ovarian cancer cells (Figure 7). Flow cytometry analysis showed that *TRPM2* plays a crucial role in the cell cycle of ovarian cancer cells and inhibits apoptosis (Figure 7).

Ovarian cancer is a highly malignant tumor characterized by a poor prognosis for patients. One of the crucial factors contributing to this outcome is the occurrence of distant metastasis during the initial stages of ovarian cancer cell growth [38]. These metastatic lesions typically manifest as tiny infiltrative metastases, making complete eradication through surgery challenging. The staging and prognosis of ovarian cancer closely correlate with the extent of cancer cell dissemination within the pelvic and abdominal cavities, as well as distant metastasis, while the size of the primary ovarian tumor plays a secondary role. In clinical practice, some patients present with relatively small primary ovarian tumors but have already developed distant metastases. Such cases often experience unfavorable prognoses, as achieving R0 resection becomes arduous during surgical intervention, and rapid postoperative recurrence is common. The results of bioinformatics research revealed a significant association between decreased expression of *TRPM2* and improved prognosis in ovarian cancer patients (Figure 7A). Additionally, in vitro experiments demonstrated that although knocking out the *TRPM2* gene did not affect the proliferation of SKOV3 cells, it diminished their invasive and migratory capabilities. These findings indicate that the *TRPM2* gene may influence the metastasis of ovarian cancer and subsequently impact patient prognosis.

Moreover, we chose the IMvigor210 cohort of urothelial cancer from Mariathasan et al. as an immunotherapeutic dataset to further validate the prognostic significance of using *TRPM2* as a biomarker for immune checkpoint blockade (ICB) therapies, such as anti-PD-L1 immunotherapy [37]. The results indicated that *TRPM2* exhibited significantly higher expression in the immune-infiltrated phenotype than in the immune-deserted phenotype, suggesting that *TRPM2* is potentially involved in the immune response of tumor tissues (Figure 9) and may be a promising target for future immunotherapy. Therefore, we claim that *TRPM2* has a promoting effect on the development and metastasis of ovarian cancer and could be a potential therapeutic target.

However, there are still limitations in the study. This study adopted a retrospective experimental design. To partially compensate for the limitations of this study, two independent datasets were employed to verify the outcomes. It is still necessary to verify the reliability of the prognostic risk model by using prospective clinical studies based on a large sample size. Furthermore, further research and exploration are needed to investigate the exact role of *TRPM2* in the pathogenesis of ovarian cancer and its potential therapeutic value.

## 4. Materials and Methods

### 4.1. Data Collection

A brief flowchart of our study is shown in Figure 1. We collected the gene sequencing data and clinical features of three sets (TCGA-OV, GSE32062, and GSE140082) from the TCGA database and GEO database. Then, 1350 ERGs with relevance scores > 5 from GeneCards (https://www.genecards.org/, accessed on 28 December 2022) were included in this study. DEGs in EOC tissues from the TCGA database and normal controls from the GTEx database were analyzed by the R software “limma” package (|log2(FC)| > 1 and *p* < 0.01) on the website (http://gepia.cancer-pku.cn/, accessed on 28 December 2022) ext, we integrated the lists of ERGs and DEGs, obtaining DEERGs.

### 4.2. Development of an ER Stress-Related Classifier Related to Prognosis

First, univariate regression analysis was conducted using the R package to screen DEERGs with significant prognostic value (*p* < 0.05). Then, the selected DEERGs were subsumed into LASSO regression, which was used to minimize the overfitting impact of the signature. LASSO regression was illustrated by the “glmnet” package in R. Finally, multivariate analysis was used to construct the ER stress-related classifier. The prognostic gene classifier was established based on the linear combination of the coefficient of Cox regression multiplied by the expression level of the gene. The prognostic gene classifier equation was as follows: risk score = (coefficient 1 × expression level of Gene 1) + (coefficient 2 × expression level of Gene 2) + (coefficient N × expression level of Gene N). The patients in the training and validation cohorts were classified into high-risk and low-risk groups according to the risk scores.

### 4.3. Survival Analysis

Kaplan–Meier survival curves were drawn using the “survival” package in R to evaluate the survival rates in the high-risk and low-risk groups. We used the ROC curve and the area under the curve (AUC) over time to evaluate the predictive value in predicting survival at one, three, five, and seven years. The risk map was illustrated by the “heatmap” package in R. Excluding samples without survival information, we retained a total of 378 EOC patients in the TCGA database. We assessed the overall survival (OS) of EOC patients by Kaplan–Meier survival analysis and calculated the *p* value using the log-rank test.

### 4.4. Determination of a Nomogram

A nomogram was drawn by the “rms” package in R, which integrates the various clinical traits and risk scores in the TCGA dataset. Calibration curves were used to evaluate the consistency of the predicted and actual survival outcomes.

### 4.5. Levels of Infiltration of Immune Cells

The CIBERSORT algorithm (permutation counts: 1000; threshold: *p* < 0.05) was used to evaluate the differences in immune infiltration levels of 22 immune cell types in the high-risk and low-risk groups [39]. We selected 10 potential inhibitory immune checkpoint genes, and the “edgeR” package in R was used to study the differential expression between the two groups.

### 4.6. qPCR Assay

The human ovarian epithelial cell line (IOSE) (as a control) and three different ovarian cancer cell lines (A2780, HEY, SKOV3) were used in our study. Cells were cultured in RPMI 1640 medium with 10% FBS and 1% penicillin–streptomycin (Gibco, Waltham, MA, USA) at 37 °C with 5% CO_2_. Experiments were performed with cells in the logarithmic growth period. We designed and constructed primers for partial selected genes and checked the sequence specificity by using BLAST (NCBI) (Table 1). GAPDH was used as a housekeeping gene, and the expression of the genes was quantified by the ΔΔCt method after correction for PCR efficiency.

### 4.7. Cell Line

The SKOV3 human cell line was purchased from Jinyuan Biological Technology Inc. (JY147; Shanghai, China). The cells were maintained in RPMI-1640 medium supplemented with 10% FBS and 1% penicillin–streptomycin (Gibco, Waltham, MA, USA) at 37 °C in a 5% CO_2_ atmosphere. All experiments were conducted using cells in the logarithmic growth phase.

### 4.8. shRNA Knockdown of TRPM2 In Vitro

To achieve shRNA knockdown of *TRPM2* in vitro, sh-*TRPM2* and negative control (sh-NC) oligonucleotides were obtained from RiboBio (RiboBio, Guangzhou, China). The specific sequences for si-PRRX1 were as follows: Si1: GATCCGGACAAGCTCTGTCTGCAAATTTCAAGA GAATTTGCAGACAGAGCTTGTCCTTTTTTG; Si2: GATCCGACAAGCTCTGTCTG CAAATCTTCAAGAGAGATTTGCAGACAGAGCTTGTCTTTTTTG; Si3: GATCCGCAAA TCTAGACAGTTCCTGCTTCAAGAGAGCAGGAACTGTCTAGATTTGCTTTTTTG. Stable transfection of the shRNAs was performed by lentiviral transfection using PEI (Polyscience, Niles, IL, USA) according to the manufacturer’s protocol.

### 4.9. CCK8 Cell Proliferation Assay

In the CCK8 assay, a 96-well plate was seeded with 2000 cells/well, and six replicate wells were included along with blank controls. The next day was designated as the 0 h time point. At 0 h, 24 h, 48 h, and 72 h, 10 μL of CCK8 reagent was added to each well, followed by incubation for 30 min. After incubation, the absorbance values were measured at 450 nm.

### 4.10. Cell Cycle Assay

Six-well plates were seeded with 1 × 10^5^ cells per well. The next day, the cells were collected and washed twice with PBS. After that, they were fixed overnight in 75% ethanol. On the following day, the cells were washed twice with PBS and stained with propidium iodide for 30 min before being analyzed using a flow cytometer.

### 4.11. Cell Migration and Invasion Assays

The migration and invasion capabilities of SKOV3 cells were evaluated using two different assays. The migration of UM cells was evaluated using the wound healing scratch assay. SKOV3 cells were seeded in six-well plates at a density of 5 × 10^5^ cells per well and allowed to reach full confluence. After serum starvation overnight, an artificial scratch wound was created at the center of each well. The scratch areas were photographed at 0, 24, and 48 h. The migration index was calculated using the following formula: Migration Index = [(original scratch width-scratch width at 24/48 h)/(original scratch width)] × 100%. The assay was repeated three times to ensure accuracy and reproducibility.

To assess cell invasion, transwell invasion assays were performed. Twenty-four-well Millicell hanging cell culture inserts with a pore size of 8.0 µm (Millipore, Billerica, MA, USA) coated with Matrigel (BD Biosciences, San Jose, CA, USA) were used as per the manufacturer’s protocol. Approximately 2 × 10^5^ cells in serum-free medium were added to the upper chamber, while the lower chamber contained 500 µL of complete medium with 10% FBS as a chemoattractant. After 48 h of incubation, cells that had invaded through the Matrigel were fixed and stained with 0.1% crystal violet for counting. Three separate fields were counted for each filter under a microscope. The assay was also repeated three times to ensure the reliability of the results.

### 4.12. ROS Measurements

For the detection of intracellular ROS levels, the H2DCFDA ROS-sensitive probe (MCE, HY-D0940) was used. SKOV3 cell lines were incubated with 5 µM staining solution in PBS in the dark at 37 °C, and confocal laser scanning microscopy (FV1000, Olympus Company, Tokyo, Japan) was used to detect the intracellular ROS generation during incubation for 1 min, 10 min, and 30 min.

### 4.13. Statistical Analysis

R software (4.2.1) was applied for statistical analysis. The qualitative variables were analyzed by Pearson’s χ^2^ test and Fisher’s exact test. A *p* value < 0.05 was considered statistically significant. Ingenuity pathway analysis (IPA) software (Qiagen, Hilden, Germany) was used to analyze the RNA-seq data to find the signaling pathways significantly affected by the regulation of ERGs in ovarian cancer and to find the key regulatory upstream factors of the differentially expressed genes.

## 5. Conclusions

In this study, we successfully generated a new ER stress-related prognostic risk classifier that yielded good accuracy in patient datasets. We further constructed a nomograph that combined the prognostic risk model and clinical parameters and could provide a good prediction of the survival outcome of EOC patients. In addition, *TRPM2* could be a potential therapeutic target of ovarian cancer cells. In total, our outcomes provide new insight into the identification of novel prognostic biomarkers and the development of therapeutic strategies for EOC.

## Figures and Tables

**Figure 1 ijms-24-14010-f001:**
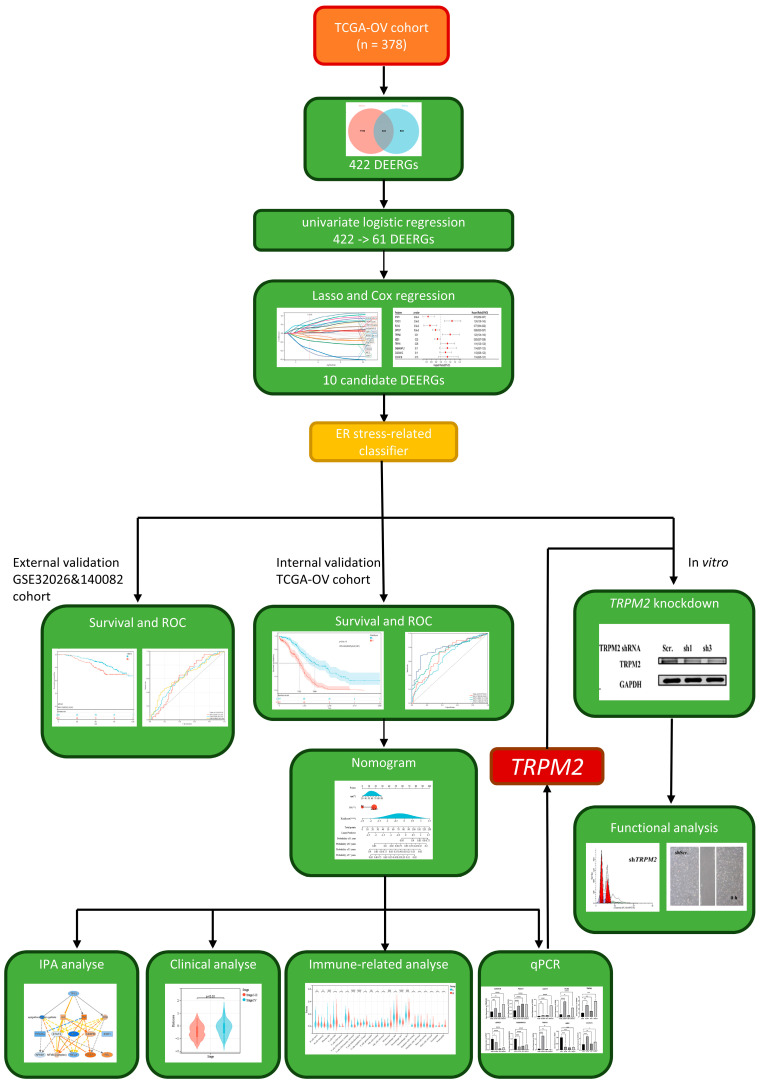
Study flow diagram.

**Figure 2 ijms-24-14010-f002:**
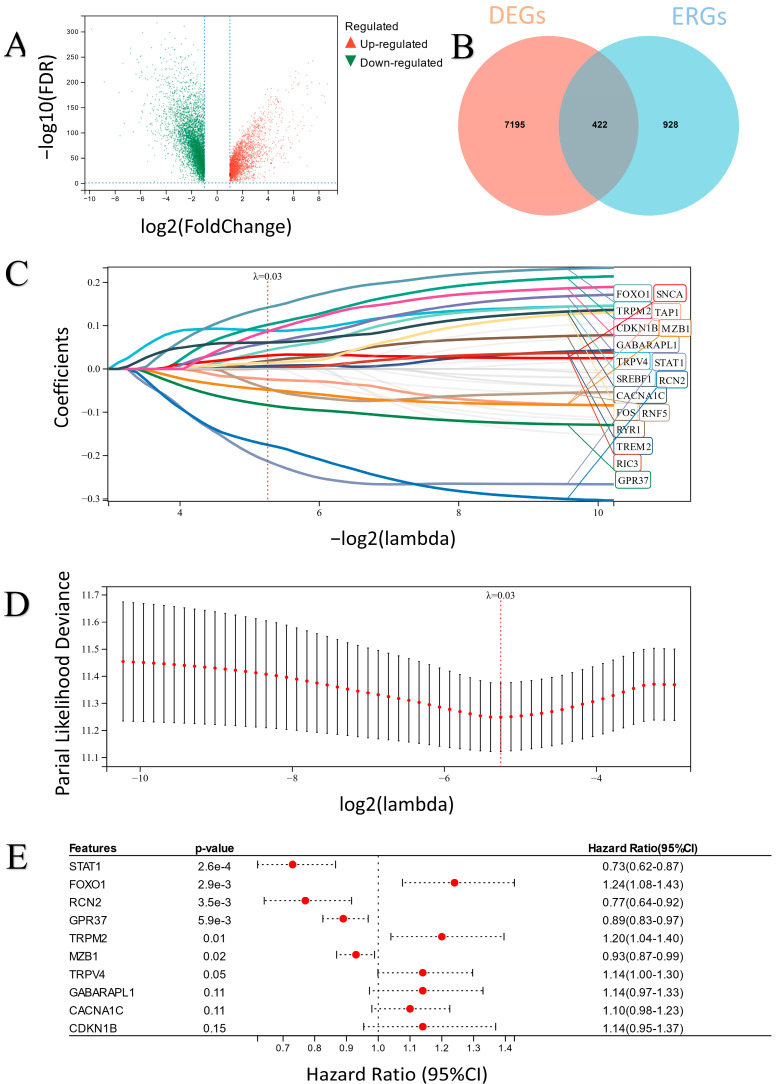
(**A**) A total of 7617 DEGs in ovarian tumor samples. (**B**) Venn plot of DEGs and ERGs. (**C**,**D**) Lasso regression method according to the survival of patients. (**E**) Multiple stepwise regression of 10 candidate DEERGs.

**Figure 3 ijms-24-14010-f003:**
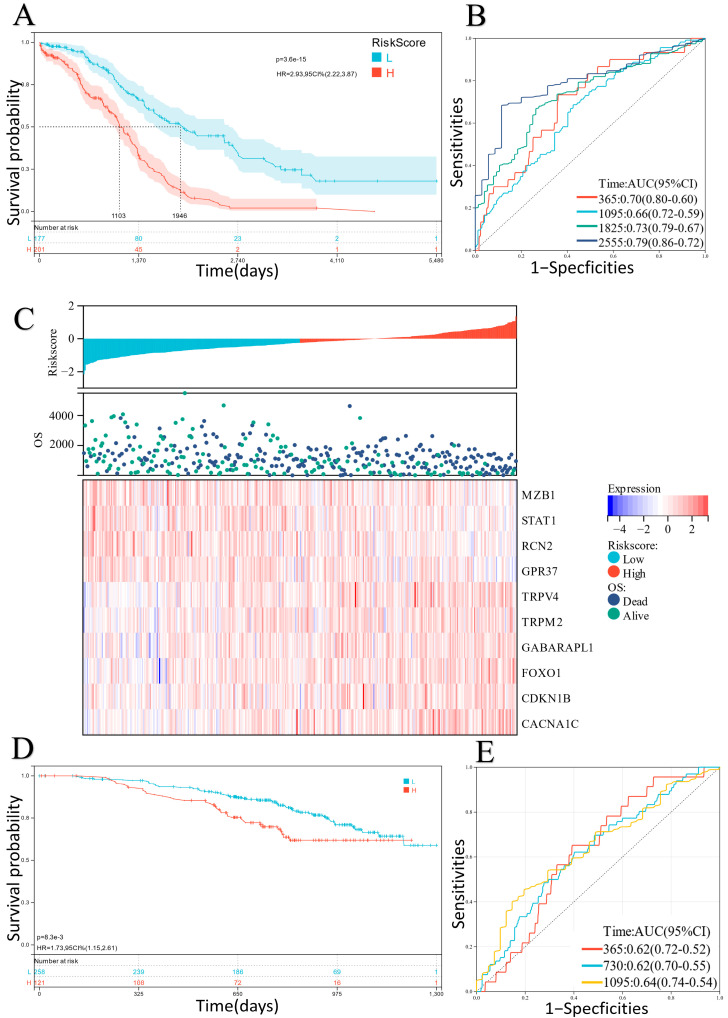
(**A**) Survival analysis of the risk scores in the TCGA dataset (H: High risk score; L: Low risk score). (**B**) ROC analysis of the risk scores in the TCGA dataset. (**C**) Layout of risk scores, survival status, and expression level of 10 candidate genes in the TCGA dataset. (**D**,**E**) Similar results were verified in the GSE32062 dataset (H: High risk score; L: Low risk score).

**Figure 4 ijms-24-14010-f004:**
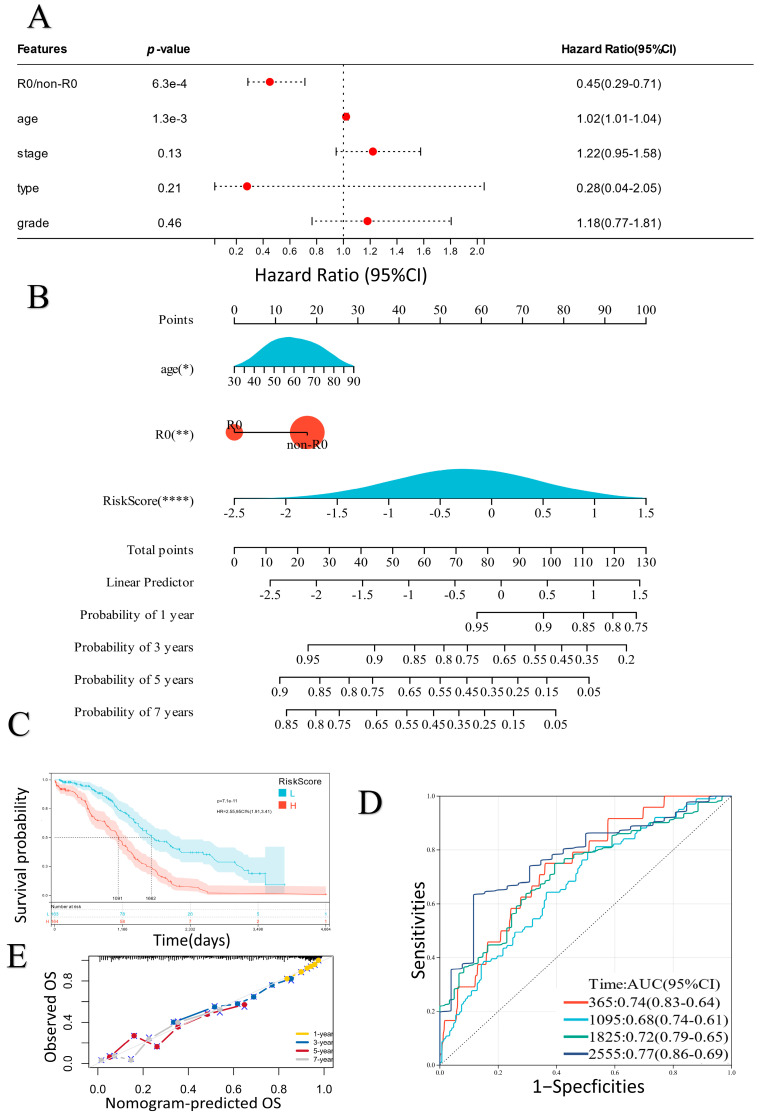
(**A**) Multivariate analysis of clinical characteristics in the TCGA dataset. (**B**) Construction of a nomogram (* *p* < 0.05; ** *p* < 0.01; **** *p* < 0.0001). (**C**) Survival analysis of the risk scores of the nomogram (H: High risk score; L: Low risk score). (**D**,**E**) ROC and calibration curves of the risk scores of the nomogram.

**Figure 5 ijms-24-14010-f005:**
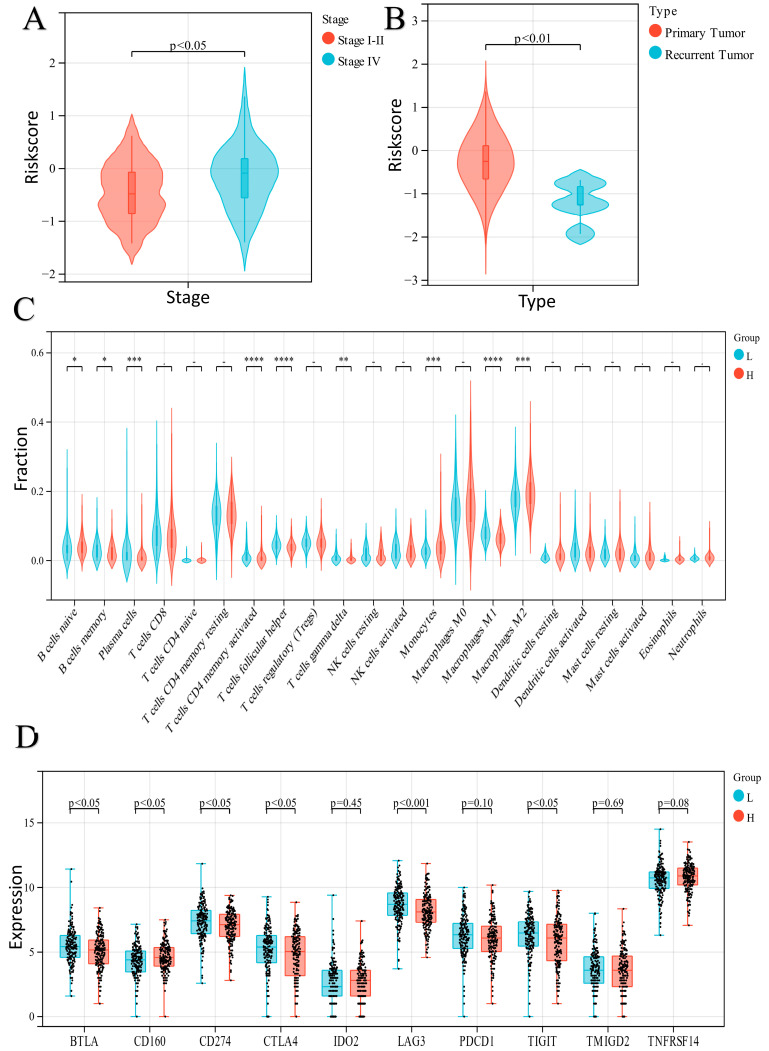
(**A**,**B**) The differences in risk score levels based on several clinical parameters. (**C**) The immune infiltration levels of the 22 cell types (H: High risk score; L: Low risk score; * *p* < 0.05; ** *p* < 0.01; *** *p* < 0.001; **** *p* < 0.0001). (**D**) The expression level of 10 immunosuppressive checkpoints (H: High risk score; L: Low risk score).

**Figure 6 ijms-24-14010-f006:**
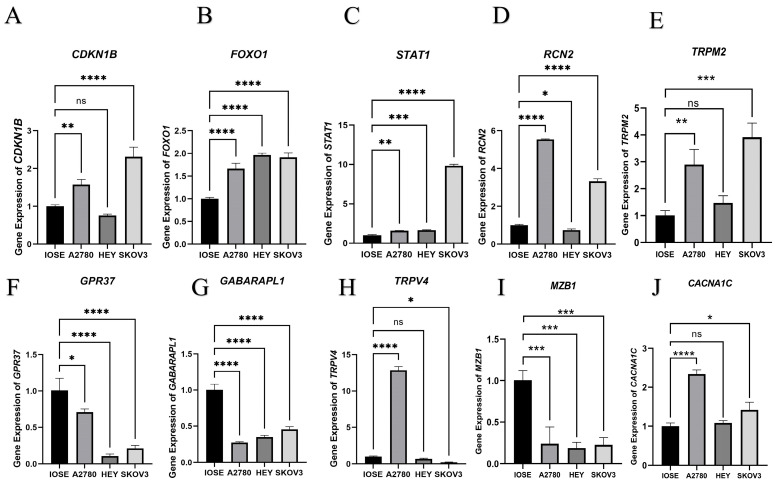
The outcome of qPCR assay in cell lines (IOSE, A2780, HEY and SKOV3) for 10 candidate DEERGs ((**A**): *CDKN1B*; (**B**): *FOXO1*; (**C**): *STAT1*; (**D**): *RCN2*; (**E**): *TRPM2*; (**F**): *GPR37*; (**G**): *GABARAPL1*; (**H**): *TRPV4*; (**I**): *MZB1*; (**J**): *CACNA1C*) among the generated ER stress-related classifiers. Error bars indicate standard errors, ns *p* > 0.05, * means *p* < 0.05, ** means *p* < 0.01, *** means *p* < 0.001, **** means *p* < 0.0001 versus the control group.

**Figure 7 ijms-24-14010-f007:**
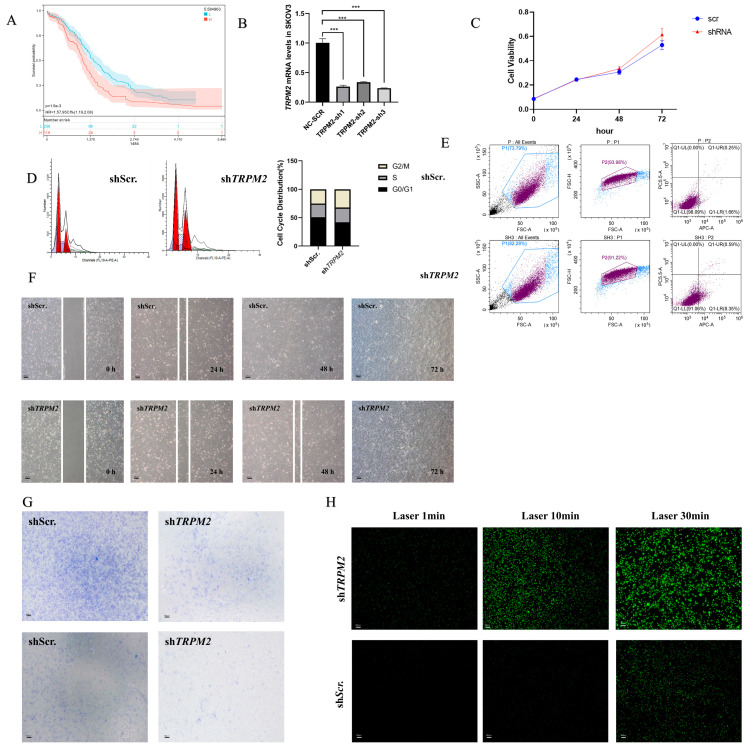
Loss-of-function experiments to silence *TRPM2* in the human SKOV3 cell line. (**A**) The survival curve of ovarian cancer patients grouped by *TRPM2* expression (H: High risk score; L: Low risk score). (**B**) Quantitative detection results of three different sh*TRPM2* mRNA levels in the SKOV3 cell line (*** indicates *p* < 0.01). (**C**) The CCK-8 assay showed that knocking down *TRPM2* had no significant effect on the proliferation of SKOV3 cells. (**D**) Specific knockdown of *TRPM2* resulted in cell cycle arrest at the G2/M phase in SKOV3 cells. (**E**) Specific knockdown of *TRPM2* significantly increased the apoptosis rate of SKOV3 cells. (**F**) The scratch assay results demonstrated that knockdown of *TRPM2* significantly reduced the migration ability of SKOV3 cells. (**G**) Transwell cell migration experiments indicated that knockdown of the *TRPM2* gene significantly reduced the migration ability (above) and invasive ability (below) of SKOV3 cells. (**H**) The ROS measurement of SKOV3 cells.

**Figure 8 ijms-24-14010-f008:**
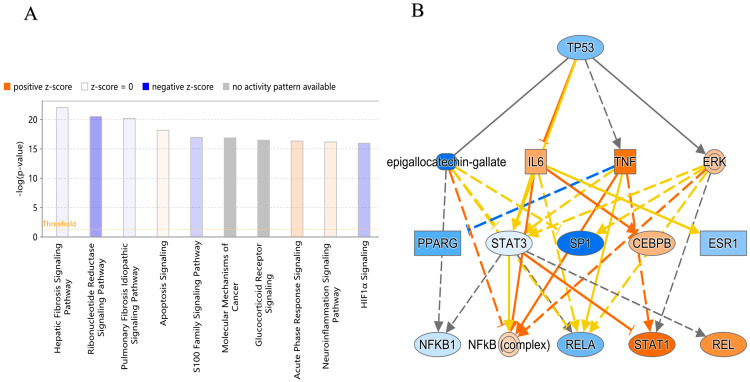
(**A**) The potential signaling pathways that were significantly affected by ER stress-related genes. In this figure, the height of the histogram shows the significance of this pathway, and the higher the column, the higher the significance. The orange column indicates that the pathway is predicted to be activated, the blue column indicates that the pathway is predicted to be inhibited, the gray column indicates that the pathway cannot be predicted temporarily, and the white column indicates that it is not predicted to be activated or inhibited. (**B**) The signaling pathways of the potential upstream regulatory factor, TP53. In this figure, the genes in orange grids are predicted to be activated and the genes in blue are predicted to be inhibited. The orange line represents activation of downstream molecules, the blue line represents inhibition of downstream molecules, and the yellow line shows inconsistent relationships with downstream molecules.

**Figure 9 ijms-24-14010-f009:**
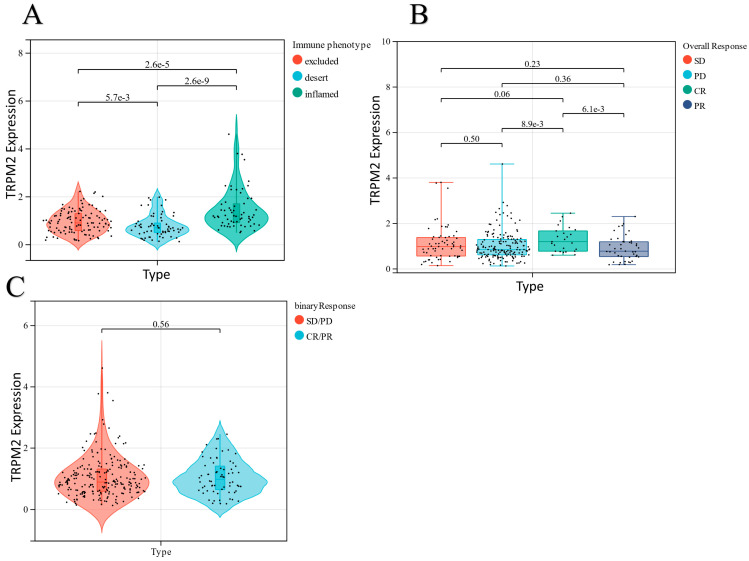
*TRPM2* in the role of anti-PD-L1 immunotherapy. (**A**) *TRPM2* expression levels were compared among distinct tumor immune phenotypes in the IMvigor210 cohort. (**B**) Distribution of *TRPM2* in distinct anti-PD-L1 clinical response groups. (**C**) The correlation of *TRPM2* with immunotherapy response. SD, stable disease; PD, progressive disease; CR, complete response; PR, partial response.

**Table 1 ijms-24-14010-t001:** The sequences of designed primers for 10 candidate DEERGs.

Gene		Sequence (5′- > 3′)
*FOXO1*	F primer	TCGTCATAATCTGTCCCTACACA
	R primer	CGGCTTCGGCTCTTAGCAAA
*TRPM2*	F primer	TCCCCGCCGAGTACATACTG
	R primer	GTCTGCTCCGATATGAACTTCTC
*RCN2*	F primer	TTCAGGTCCCGGTTTGAGTCT
	R primer	TCAAGCCTGCCATCGTTATCT
*MZB1*	F primer	AGTTGGTCTACACGGATGTCC
	R primer	CTTGGTCCACTTCTCGAACTC
*GPR37*	F primer	ATGTCGCGGCTACTGCTTC
	R primer	GCAGAACGTCTCTTGCAGAAT
*GABARAPL1*	F primer	ATGAAGTTCCAGTACAAGGAGGA
	R primer	GCTTTTGGAGCCTTCTCTACAAT
*CDKN1B*	F primer	ATCACAAACCCCTAGAGGGCA
	R primer	GGGTCTGTAGTAGAACTCGGG
*STAT1*	F primer	CAGCTTGACTCAAAATTCCTGGA
	R primer	TGAAGATTACGCTTGCTTTTCCT
*TRPV4*	F primer	TCCACCCTATATGAGTCCTCGG
	R primer	TAGGTGCCGTAGTCAAACAGT
*CACNA1C*	F primer	TGATTCCAACGCCACCAATTC
	R primer	GAGGAGTCCATAGGCGATTACT

## Data Availability

The datasets analyzed during the current study are available in the TCGA-OV dataset, https://portal.gdc.cancer.gov/ (accessed on 28 December 2022), GSE32062 dataset, GSE32062 [Accession]-GEO DataSets-NCBI (nih.gov), and GSE140082 dataset, GSE140082 [Accession]-GEO DataSets-NCBI (nih.gov).

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
