# Peer review of "Endoplasmic Reticulum Stress-Related Ten-Biomarker Risk Classifier for Survival Evaluation in Epithelial Ovarian Cancer and TRPM2: A Potential Therapeutic Target of Ovarian Cancer"

_ijms, 2023, doi:10.3390/ijms241814010_

Round 1

Reviewer 1 Report

This is an interesting and highly relevant paper addressing the challenge of early diagnosis of a lethal gynecological cancer through potential biomarker assessment. 

While various aspects of the research require validation and further analyses, the authors presented interesting results, tackled from various methodical angles.  This lays an important foundation for future research.

I believe this manuscript would be of interest to researchers in the gyneco-oncological field. 

Minor editing of the English would improve the manuscript, specifically in the improvement of more formal writing- e.g. removing colloquial terms such as 'Last but not least...'

Author Response

Dear reviewer,

We thank the reviewer for taking the time out of the busy schedule to review my manuscript and provide sincere suggestions and expectations. Early diagnosis of ovarian cancer has always been a clinical practice challenge, which generates relative important impacts patient prognosis. we hope that our research would contribute to the early detection of gynecological tumors.

We thank the reviewer for the relevant remark, indeed, we have only conducted preliminary phenotypic functional validation of the TRPM2 gene, and there are many aspects that need further expansion and verification, for instance the phenotypic and mechanistic studies of the other nine identified genes. We will explore these areas in our future work.

Regarding to the language, we thank the reviewer for the valuable comments. We have thoroughly checked and revised the language carefully. We follow the suggestion from the reviewer and removed the colloquial expressions to make the manuscript more formal for publication.

Reviewer 2 Report

In the manuscript by Zhang et al the data analysis part has been performed well though the images need to be improved by increasing the front size of the text in it. While in the validation part the shRNA mediated silencing of the TRPM2 has no effect on the cell proliferation, marginal effect on apoptosis and cell cycle but significant effect on the migratory potential of the ovarian cancer cells. While the manuscript started with a goal to identify Biomarker Risk Classifiers which impart was done by In Silico analysis but the invitro data does not support the In Silico research findings. Regarding the TRPM2, it is known to plays an important role in mediating cell death induced by miscellaneous oxidative stress-inducing pathological factors, but the authors have not evaluated the ROS levels in shTRPM2 cells.

Author Response

Dear reviewer:

First of all, we would like to thank the reviewer for your time and valuable comments, which made it possible to improve the quality of the manuscript. We really appreciate all your constructive comments and suggestions on our manuscript. We have taken the comments into consideration and tried our best to improve the manuscript.

1) We thank the reviewer for the valuable suggestion on image visualization, and we have increased the front size of the text in the images.

2) We thank the reviewer for this comment. Indeed, the in vitro data did not fully support the in silico research findings. We consider this to be a very precise and constructive question, and it is crucial for us to engage in further and deeper discussion on this topic. To make it clearer, we have done the experiment about ROS assay and put the result in Figure 7I. We rephrased our original sentence and added the following discussion (line 315-331):

Ovarian cancer is a highly malignant tumor characterized by a poor prognosis for patients. One of the crucial factors contributing to this outcome is the occurrence of distant metastasis during the initial stages of ovarian cancer cell growth[1]. These metastatic lesions typically manifest as tiny infiltrative metastases, making complete eradication through surgery challenging. The staging and prognosis of ovarian cancer closely correlate with the extent of cancer cell dissemination within the pelvic and abdominal cavities, as well as distant metastasis, while the size of the primary ovarian tumor plays a secondary role. In clinical practice, some patients present with relatively small primary ovarian tumors but have already developed distant metastases. Such cases often experience unfavorable prognoses, as achieving R0 resection becomes arduous during surgical intervention, and rapid postoperative recurrence is common. Our bioinformatics research had discovered a significant association between decreased expression of TRPM2 and improved prognosis in ovarian cancer patients. Additionally, in vitro experiments had demonstrated that although knocking out the TRPM2 gene did not affect the proliferation of SKOV3 cells, it diminished their invasive and migratory capabilities. These findings indicated that the TRPM2 gene may influence the metastasis of ovarian cancer and subsequently impact patient prognosis. The above discussion content has been added to the manuscript to enrich the paper.

As the reviewer mentioned, TRPM2 plays an important role in mediating cell death induced by miscellaneous oxidative stress-inducing pathological factors. It is necessary to evaluate the ROS levels in shTRPM2 cells. We used ROS-sensitive probe H2DCFDA to detect the intracellular ROS generation. The downregulation of TRPM2 levels resulted in an elevation in the intracellular reactive oxygen species (ROS) levels, indicating that TRPM2 may have enhanced ovarian cancer cell activity by inhibiting the ROS. We have put the remark in the discussion to enrich the paper. This result is also similar to the conclusions of other studies[2]. The above contents related to ROS measurements has been added to the manuscript.

  1. Sommerfeld L, Finkernagel F, Jansen JM, Wagner U, Nist A, Stiewe T, Müller-Brüsselbach S, Sokol AM, Graumann J, Reinartz S et al: The multicellular signalling network of ovarian cancer metastases. Clin Transl Med 2021, 11(11):e633.
  2. Bao L, Festa F, Freet CS, Lee JP, Hirschler-Laszkiewicz IM, Chen SJ, Keefer KA, Wang HG, Patterson AD, Cheung JY et al: The Human Transient Receptor Potential Melastatin 2 Ion Channel Modulates ROS Through Nrf2. Sci Rep 2019, 9(1):14132.

Reviewer 3 Report

The article as a whole is of scientific interest, contains new, evidence-based and statistically very representative material.

However, there are some errors and technical shortcomings in the text that need to be corrected, namely:

1) Abbreviations  DEGs and DEERGs used on page 2 and henceforth are introduced only on page 12

2) on page 3 we are talking about figure 1E, this part is not in figure 1

3) In figures 3B, 4 C and D, on all parts of figures 5 and 9 there is a very small and almost unreadable font.

4) In the caption to figure 5, it is necessary to explain what subgroups H and L mean.

Author Response

Dear reviewer:

We thank the reviewer for taking the precise time to review our manuscript and provide sincere suggestions. We sincerely appreciate the reviewer’s feedback on the problems with the images in our manuscript. The valuable suggestions have helped us identify numerous formatting errors in the images and have made our manuscript more readable. Below, we will provide a point-by-point response to your suggestions:

  • Abbreviations  DEGs and DEERGs used on page 2 and henceforth are introduced only on page 12. Due to the initial variation in the structure and order of the draft, this error occurred. We have now rectified it by including the full names of DEGs and DEERGs at their first appearance (line 70- 72).
  • On page 3 we are talking about figure 1E, this part is not in figure 1. This content is located in Figure 2E, and we have already made correction to the manuscript. (line 83).
  • In figures 3B, 4 C and D, on all parts of figures 5 and 9 there is a very small and almost unreadable font. In addition to the mentioned image, we have also enhanced all the images throughout the entire manuscript by increasing the font size of the text within the images. We have made our best effort to improve the readability of the article.
  • In the caption to figure 5, it is necessary to explain what subgroups H and L mean. We have reviewed all the images in the article and have now included labels indicating the meaning of "H" and "L" in the images (H: high riskscore; L: low riskscore).

Round 2

Reviewer 2 Report

The authors have incorporated the required changes to the manuscript.